# Nitrous Oxide Adsorption and Decomposition on Zeolites and Zeolite-like Materials

**DOI:** 10.3390/molecules27020398

**Published:** 2022-01-08

**Authors:** Leonid M. Kustov, Sergey F. Dunaev, Alexander L. Kustov

**Affiliations:** 1Chemistry Department, Moscow State University, 1 Leninskie Gory, Bldg. 3, 119991 Moscow, Russia; dunaev@general.chem.msu.ru (S.F.D.); kyst@list.ru (A.L.K.); 2N.D. Zelinsky Institute of Organic Chemistry RAS, 47 Leninsky Prosp., 119991 Moscow, Russia; 3Institute of Ecotechnologies and Engineering, National University of Science and Technology MISiS, 4 Leninsky Prosp., 119049 Moscow, Russia

**Keywords:** HZSM-5 zeolite, N_2_O decomposition, titanosilicalites, Lewis acid sites, diffuse-reflectance IR spectroscopy

## Abstract

Decomposition of N_2_O on modified zeolites, crystalline titanosilicalites, and related amorphous systems is studied by the catalytic and spectroscopic methods. Zinc-containing HZSM-5 zeolites and titanosilicalites with moderate Ti/Si ratios are shown to exhibit a better catalytic performance in N_2_O decomposition as compared with conventionally used Cu/HZSM-5 zeolites and amorphous Cu-containing catalysts. Dehydroxylation of the HZSM-5 zeolite by calcination at 1120 K results in an enhancement of the N_2_O conversion. The mechanism of the reaction and the role of coordinatively unsaturated cations and Lewis acid sites in N_2_O decomposition are discussed on the basis of the spectroscopic data.

## 1. Introduction

The problem of N_2_O decomposition remains to gradually attract attention in view of the development of green technologies. This problem is related to the synthesis of adipic acid, which yields N_2_O as a side product, as well as NO_x_ abatement in exhaust gases of power plants or waste anesthetic gas purification. Furthermore, the reaction of N_2_O decomposition was shown to be the initial and key stage in the processes of selective oxidation of aromatic compounds with N_2_O under mild conditions using zeolites as catalysts [1,2,3,4,5,6]. It was shown that coordinatively unsaturated cations (iron species and framework Lewis acid sites) are responsible for the catalytic activity of dehydroxylated HZSM-5 zeolites both in N_2_O decomposition and in the reactions of oxidation of various aromatic substrates using N_2_O [3,4,5]. In the art, oxide systems are known as N_2_O decomposition catalysts, with amorphous copper oxide, for instance, Cu-Me/Al_2_O_3_, as well as cobalt oxide systems [7], magnesium cobaltite Mg*_x_*Co_1−*x*_Co_2_O_4_ [8], CoO_x_-CeO_2_ [9] or Co-Ce spinel [10] as quite active, although most of the Co-based catalysts, except for the Co-Ce spinel, demonstrate high conversion only at high temperatures (870–1070 K). Ceria-zirconia behaves nearly similar to Co-oxide materials [11].

Other supported catalysts, such as rhodium on lanthanum silicate Rh/La_10_Si_6−*x*_Fe*_x_*O_27−δ_ or Pt/ZrO_2_ providing a 100% N_2_O conversion at temperatures as high as 870 K [12,13] or Pt, Ir, and Pd supported on Al_2_O_3_ [14] have been also studied. However, the use of noble metals seems to be an expensive way to N_2_O abatement. Furthermore, carbon nanotubes were predicted by DFT calculations to catalyze this reaction [15].

Among the most efficient catalysts used for N_2_O decomposition, high-silica zeolites modified with iron [5,16], rhodium [17], copper [18], ruthenium [19], and mixed Co-In [20] ions were shown to demonstrate the best performance. The reported catalysts provide a complete conversion of nitrous oxide to nitrogen and oxygen at 620 K. Ru(0) nanoclusters prepared by the reduction of Ru(III) ions, as well as osmium(III) species were found to be less active compared with ruthenium ions. The systems containing Fe, Cu, Co, and Ru metal ions exhibited a much better catalytic performance in N_2_O decomposition as compared with other modified and non-modified zeolites [21], as well as other amorphous oxide systems [22,23]. The main disadvantages revealed, for example, by the Cu-catalysts for N_2_O decomposition are their low thermal stability (they irreversibly lose the activity after overheating to T > 870 K) and poor tolerance to admixtures of H_2_O, CO, CO_2_, and hydrocarbons, which are present in real gas mixtures and act as poisons. The behavior of catalysts definitely depends on the presence of water vapor in the feed, as well as other residual components (NO, O_2_, NO_2_) that may interfere with the N_2_O decomposition process [24]. However, in the mixture, we will limit our scope with the model conditions, without the introduction of other potentially important ingredients.

The aim of this work was to find new zeolite and zeolite-like catalysts that are active in N_2_O decomposition and to study the nature of active sites and plausible reaction mechanisms, with an emphasis on the role of coordinatively unsaturated cations. Three groups of catalysts were chosen for the investigation:−Dehydroxylated HZSM-5 zeolites and ZSM-5 zeolites modified with zinc oxide, which have been studied earlier from the point of view of the nature and strength of Lewis acid sites [25,26], i.e., the systems containing strong coordinatively unsaturated cations (Lewis acid sites);−Crystalline Ti-silicalites that are widely used as efficient catalysts for the selective oxidation of phenol into diphenols by H_2_O_2_ in the liquid phase [27];−Amorphous catalysts, based on the Ti/SiO_2_ system, which differ in the Ti/Si ratio and in the preparation method.

For comparative purposes, the well-known Cu-ZSM-5-type catalysts for N_2_O decomposition, as well as amorphous Cu-containing oxide systems were also studied.

## 2. Results and Discussion

To evaluate the relative strength of coordinatively unsaturated cations in the modified zeolites under study and to rank the samples according to the relative concentration of strong electron-acceptor centers, IR spectra of molecular hydrogen, as a probe for low-coordinated cations [28], were measured.

Figure 1 shows the IR spectra of H_2_ adsorbed at 77 K on three representative samples containing rather strong coordinatively unsaturated cations: (1) Dehydroxylated HZSM-5 zeolite, (2) Cu/HZSM-5, and (3) Zn/HZSM-5. The absorption bands in the region of 4100–4120 cm^−1^ correspond to weakly bonded H_2_ complexes with bridging Si(OH)Al and terminal SiOH groups, respectively [28], whereas the bands below 4100 cm^−1^ are shown [28] to belong to complexes of molecular hydrogen with coordinatively unsaturated cations (or Lewis acid sites) that exhibit electron-accepting properties. The stronger the interaction in the complex, i.e., the stronger the electron-acceptor center, the larger the shift of the corresponding band of adsorbed H_2_ toward lower frequencies measured relative to the frequency of the H-H stretching vibration in the gas phase (ν_H_-_H_ = 4163 cm^−1^) [28]. As seen from the spectra shown in Figure 1, the strongest coordinatively unsaturated cations are present in the Zn/ZSM-5 zeolite (ν_H-H_ = 3955, 4010 and 4070 cm^−1^, for these bands Δν_H_-_H_ = 208, 153, and 93 cm^−1^, respectively), whereas the weakest centers among the three catalysts under consideration are revealed in the dehydroxylated HZSM-5 zeolite (ν_H_-_H_ = 4010 and 4035 cm^−1^, Δν_H_-_H_ = 153 and 128 cm^−1^, respectively). The Cu/ZSM-5 zeolite, which is the well-known active catalyst for N_2_O decomposition, manifests an intermediate strength of the electron-acceptor centers (ν_H_-_H_ = 3970 and 4060 cm^−1^, Δν_H_-_H_ = 193 and 103 cm^−1^, respectively). Of note, the concentration of the strongest electron-acceptor centers is the highest for the Zn/HZSM-5 zeolite. Moreover, it is noteworthy that a further increase in the loading of copper in the Cu/HZSM-5 zeolite from 1 to 3 wt%, as well as an increase in the loading of zinc in the Zn/HZSM-5 zeolite over 5 wt%, have no appreciable effect on the spectral pattern, i.e., the concentration of strong Lewis acid sites. Furthermore, this increase does not improve the catalytic performance of the Cu-zeolite and Zn-zeolite catalysts.

The pre-adsorption of a small amount of N_2_O at 300 K on the zeolite samples, which precedes the adsorption of H_2_ results in the disappearance of the low-frequency absorption bands attributed to the H_2_ complexes with low-coordinated cations. However, this has no considerable influence on the intensity of the high-frequency bands (ν_H_-_H_ = 4100–4120 cm^−1^) assigned to the complexes with OH groups. This experiment shows that adsorption of N_2_O occurs on the low-coordinated metal cations that are responsible for the appearance of the corresponding absorption bands in the IR spectra of adsorbed hydrogen.

The adsorption of N_2_O on the zeolite samples results in the appearance of the absorption bands at 2285–2230 cm^−1^ (Figure 2). The frequency of gaseous N_2_O is 2224 cm^−1^. Herein, we observe one band at 2230 cm^−1^, which is close to the gas-phase value (physically adsorbed N_2_O) and a shifted band at 2285 cm^−1^ due to complexes with zinc species (electron-acceptor centers). The largest shift of the N_2_O band with respect to the corresponding band position for N_2_O molecules in the gas phase is observed for Zn/HZSM-5 zeolites (ν = 2285 cm^−1^, Δν = 50 cm^−1^), which indicates the strongest polarization and activation of the N_2_O molecule by the electron-acceptor sites of the Zn/HZSM-5 zeolite. According to our previous spectroscopic data and quantum-chemical calculation [1,2], the N_2_O molecule is preferably adsorbed on the Lewis acid center (for instance, on trigonally coordinated aluminum ions) by a two-point mechanism, which also involves a neighboring oxygen atom of the surface cluster. In this case, adsorption of N_2_O is accompanied by a considerable change of the geometry of the molecule, in particular, by a substantial decrease of the NNO angle (from 180 to 140°) and by a strong polarization of the N-O bond, which favors the further decomposition of the N_2_O molecule with the evolution of N_2_ into the gas phase and chemisorption of atomic oxygen [1,2]. Evidently, the extent of N_2_O polarization and activation, and thus, the rate of decomposition are governed by the strength of coordinatively unsaturated cations. Correspondingly, heating of the Zn/HZSM-5 zeolite with pre-adsorbed N_2_O at 520 K for 1 h directly in the IR cell (under static conditions) leads to the complete decomposition of N_2_O, and the bands at 2285–2240 cm^−1^ vanish from the spectrum, whereas the corresponding band at 2355 cm^−1^ appears due to the molecular nitrogen formed upon the N_2_O decomposition. For comparison, heating of the dehydroxylated HZSM-5 zeolite with pre-adsorbed N_2_O at 520 K for 1 h results only in a partial decomposition of N_2_O, in accordance with a weaker strength of the low-coordinated cations (Lewis acid sites).

To reveal subtle distinctions in the properties of the modified zeolites related to N_2_O decomposition, we tested the samples in the flow catalytic unit at 620–900 K. The reaction conditions and the conversion degrees for the N_2_O decomposition on the modified HZSM-5 zeolites and some Cu-containing amorphous catalysts used for NO_x_ decomposition are summarized in Table 1. In agreement with the spectroscopic data, the dehydroxylated HZSM-5 zeolite exhibits a poor conversion even at enhanced temperatures (720 K), and the Zn-containing zeolites reveal the best performance. These catalysts are active at low temperatures as 620 K (the conversion of 85%), while the known Cu/HZSM-5 system exhibits a considerable inferior performance (the conversion does not exceed 20%) under the same conditions. Of note, both samples of the amorphous Cu-containing catalysts show a poor performance as compared with the Zn- and Cu-zeolites. The presence of low-coordinated metal ions (zinc or copper) should clearly be considered as the pre-requisite for the efficient N_2_O decomposition. Therefore, the spectroscopic and catalytic data indicate that strong electron-acceptor (low-coordinated) metal ions, which should actually be considered as Lewis acid-base pair sites containing a low-coordinated metal ion and an oxygen anion of the framework, are presumably the active centers responsible for the N_2_O decomposition on the modified zeolites. With the analogy from our previous studies and taking into account the results of quantum-chemical calculations [1,2], we may propose the following mechanism of N_2_O decomposition on strong coordinatively unsaturated metal ions, which involves strong perturbation of the N_2_O molecule and further formation of the chemisorbed oxygen atom. Here, the latter is consumed for the recombination or scavenged by the second N_2_O molecule, yielding N_2_ and O_2_ (Figure 1):

The second group of catalysts studied in the reaction of N_2_O decomposition comprised crystalline Ti-silicalites with different Si/Ti ratios and amorphous TiO_2_-SiO_2_ systems. The Ti-silicalites have been chosen for the investigation in the title reaction, since they exhibit unique catalytic properties in the reactions of selective oxidation of phenol into diphenols using H_2_O_2_ as an oxidizing agent [27,29]. The active centers responsible for these properties of the Ti-silicalites are should be titanyl groups Ti=O or isolated tetrahedral Ti^+4^ ions [29]. Accordingly, the reaction is assumed to involve Ti-OOH fragments in the coordination sphere of the isolated Ti^+4^ ions. Taking into account the fact that the N_2_O molecule contains labile oxygen, similar to the H_2_O_2_ molecule, and with due regard to the similarity of the reaction mechanisms for the selective oxidation with N_2_O and H_2_O_2_, which include activation and decomposition of the molecule of the oxidizing agent, we may assume that the catalysts which are active in the reactions involving H_2_O_2_, i.e., Ti-silicalites, will be active in the reaction of N_2_O decomposition.

Table 2 presents the results of catalytic testing of various Ti systems in the reaction of N_2_O decomposition. The N_2_O conversion for the crystalline Ti-silicalites of the TS-1 type (four samples) passes through a maximum at Si/Ti = 32. Of note, the performance of the crystalline Ti-silicalite with the Si/Ti ratio equal to 32 is higher than the known Cu/HZSM-5 catalyst, especially at low temperatures (620–645 K). The dome-shaped dependence of the N_2_O decomposition rate for the Ti-silicalites versus the Si/Ti ratio may be accounted for in the following way. Evidently, a decrease in the performance with the increasing Si/Ti ratio from 32 to 38 results from a diminution of the concentration of active isolated Ti^+4^ ions in the framework. A decrease in the conversion degree when the Ti content in the samples increases (the Si/Ti ratio decreases to 20–15), may be equally explained by a decrease in the concentration of the active isolated Ti^+4^ species as a result of the growth of the concentration of the pair Ti^+4^ centers, which are likely inactive (or less active, as compared with the isolated Ti^+4^ species). The concentration of the isolated Ti^+4^ species may also decrease due to the formation of octahedral Ti^+4^ centers, in particular, extra-framework octahedral species, for instance, in the form of anatase, which is inactive in the N_2_O decomposition, at least in the temperature range studied. The latter hypothesis is consistent with the data presented by Bellussi et al. [30], who showed that the probability of the formation of anatase (or in general, octahedral Ti^+4^ ions) during the synthesis of Ti-silicalites of the TS-1 type drastically increases, when the Si/Ti ratio approaches 20 and lower values.

For a comparison with the crystalline Ti-silicalites, we also tested the performance of amorphous TiO_2_-SiO_2_ samples. Here, it is known [31] that the amorphous Ti/SiO_2_ is completely inactive in the reaction of selective oxidation of phenol with aqueous solutions of H_2_O_2_. Moreover, these samples exhibited a very low conversion in the reaction of N_2_O decomposition. Furthermore, they revealed some N_2_O conversion only at high temperatures of 720–770 K, while the crystalline Ti-silicalites with a close Si/Ti ratio were very active at 620 K.

To ascertain the coordination state of titanium ions in the crystalline and amorphous Ti-systems, we used the XPS method. Figure 3 depicts the representative XP spectra of two crystalline samples and one amorphous catalyst. The spectra contain a characteristic line of Ti 3p_3_/_2_ in the range of the binding energies of 460.0–458.7 eV. For the Ti-silicalite with a moderate Si/Ti ratio (32), a sharp peak at 460.0 eV is observed, which is ascribed to tetrahedrally coordinated Ti^+4^ ions, whereas for Ti-silicalites with lower Si/Ti ratios (22.9 and 14.4), a superposition of the peak at 460.0 eV with the second line with the maximum at 458.7 eV is revealed as a result of the presence of octahedrally coordinated Ti^+4^ ions. A similar spectral pattern is observed for the amorphous TiO_2_-SiO_2_ sample.

With an analogy regarding the chemistry of processes based on H_2_O_2_, one may consider two plausible reaction mechanisms for N_2_O decomposition on the Ti-silicalites (Figure 2):

The first mechanism is similar to the one proposed for the modified zeolites contain-ing coordinatively unsaturated cations, such as Zn/HZSM-5, except for the fact that Ti^+4^ ions are not in the trigonal but in the tetragonal coordination. However, taking into account the fact that (1) for Ti^+4^ ions the characteristic coordination numbers are 4 and 6, and (2) tetragonal Ti^+4^ ions in the framework of Ti-silicalites are capable of coordinating additional adsorbate molecules, we may consider the first mechanism as one of the possible ways of N_2_O transformation. The aforementioned XPS data lend some support for this mechanism of N_2_O decomposition. However, of note, the presence of five-coordinated titanium ions cannot be excluded, since the XPS pattern represents a superposition of at least two, maybe three lines. The five-coordinated titanium ions are also coordinatively unsaturated and therefore, can take part in the reaction. The second mechanism involves a cyclic peroxo complex, which also seems quite probable in view of the data obtained for the so-called “reactive silica” [32]. In any case, discrimination between these two mechanisms should be done in the future research, probably, with the help of labelled isotopes of oxygen.

Figure 4 displays the IR spectra of two samples of the crystalline Ti-silicalites measured after N_2_O adsorption at room temperature and after heating the sample with pre-adsorbed N_2_O at 520 K directly in the IR cell under static conditions. Unlike Zn/HZSM-5 zeolites, the adsorption of N_2_O does not result in a considerable polarization and perturbation of the molecule, and the band position for adsorbed N_2_O (Δν_H_-_H_ = 2235–2225 cm^−1^) is very close to the physically adsorbed N_2_O. Nevertheless, heating of the samples at 570 K for 1 h results in the complete (the sample with Si/A = 32) or considerable (the sample with Si/Al = 14.4) disappearance of the N_2_O absorption bands. Simultaneously, the bands of N_2_ at 2360–2340 cm^−1^ are formed, thereby indicating the decomposition of N_2_O. These data agree fairly well with the catalytic data presented in Table 2.

## 3. Materials and Methods

The dehydroxylated HZSM-5 zeolite was prepared by calcination of the HZSM-5 zeolite (Si/Al = 20) in a vacuum at 1120 K for 2 h. Zn/HZSM-5 catalysts with ZnO loadings of 1–5 wt% were synthesized by wet impregnation of the HZSM-5 zeolite with a 1 M aqueous solution of Zn(NO_3_)_2_, with further drying at 390 K in air and successive calcination in air at 820 K for 4 h and 920 K for 4 h. Four samples of crystalline Ti-silicalites of the TS-1 type with Si/Ti ratios of 14.4, 22.9, 32.0, and 37.9 were prepared according to the known procedure [30]. The Cu/HZSM-5 zeolite with 3 wt% Cu, which corresponded to the maximum conversion on the zeolite in N_2_O decomposition, was prepared by wet impregnation of the HZSM-5 zeolite similar to the Zn/HZSM-5 samples. The crystallinity of the zeolites and Ti-silicalites under study, monitored by XRD, was close to 95–100%.

Samples of amorphous Ti/SiO_2_ catalysts were prepared by the method of chemical vapor deposition (CVD) using TiCl_4_ and the sample of silica gel with successive hydrolysis or by coprecipitation of TiO_2_ and SiO_2_. The resulting Ti/SiO_2_ catalysts were characterized by TiO_2_ loadings of 0.5–80 wt%. The Co-Cr-Cu/Al_2_O_3_ catalyst for N_2_O decomposition was prepared by the co-precipitation of equimolar amounts of Co, Cr, and Cu from their nitrate precursors, in the presence of γ-Al_2_O_3_ (surface area, 170 m^2^/g) with further calcination at 770 K for 2 h.

In this paper, the structure of all the studied zeolite samples, including the starting HZSM-5, dehydroxylated HZSM-5, Zn/HZSM-5, Cu/HZSM-5, and TS-1 samples with any Si/Ti ratio, present the same MFI type, as determined by XRD.

Prior to the catalytic tests, all of the samples were activated at 770 K for 4 h in an air flow. The catalytic reaction of N_2_O decomposition was studied in a flow setup at 620–900 K and an N_2_O + He (1:1) flow rate of 20–60 mL/min. The sample loading was 0.2 g. The catalyst (0.5–1 mm particle size) was diluted with quartz (1:1). The reaction products and unreacted N_2_O were analyzed by gas chromatography (a Krystalux chromatograph) using a 1-m Porapak Q column. The only products of N_2_O decomposition were N_2_ and O_2_. Diffuse-reflectance IR spectra were measured in the range of 4000–8000 and 2000–4000 cm^−1^ with a Beckman Acta-M-VII; and Perkin-Elmer 580 B spectrophotometer, respectively, according to the reported procedures [28]. Molecular hydrogen adsorbed at 77 K and a pressure of 30 Torr were used as a probe for coordinatively unsaturated cations [28,29]. Nitrous oxide was adsorbed on the samples at 300 K and a pressure of 10–30 Torr.

XP spectra were measured with a XSAM-800 spectrometer using the MgKa excitation. The C_1S_ line at 285.0 eV was used as a reference.

## 4. Conclusions

In conclusion, the obtained catalytic and spectroscopic data allow the arrangement of the systems under study in the following sequence, and according to their performance in N_2_O decomposition: Zn/HZSM-5 > TS-1 (III) > Cu/HZSM-5 > TS-1 (IV) > (Co, Fe)/Cr/Cu/Al_2_O_3_ > HZSM-5 > TS-1 (I), TS-1 (II), TiO_2_-SiO_2_. In addition, from these data, two new catalytic compositions, i.e., ZnO/HZSM-5 and Ti-silicalite, with a moderate Si/Ti ratio, exhibit a better performance in the reaction of N_2_O decomposition, as compared with the conventional Cu-containing zeolite and oxide catalysts. The key role played by coordinatively unsaturated Zn, Cu or Ti ions, as non-framework (Zn, Cu) or framework ions in the N_2_O decomposition has been revealed. Even the dehydroxylated HZSM-5 zeolite, containing rather strong Lewis acid sites (but still weaker than those in Zn/HZSM-5 catalysts), is more active in the reaction of N_2_O decomposition compared with the conventionally calcined HZSM-5 zeolite, which contains predominantly Bronsted acid sites. Furthermore, a considerably high N_2_O conversion reaching 85% is observed for the most active catalysts (Zn/HZSM-5) under rather mild reaction conditions (T = 620 K).

## Data Availability

Not applicable.

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
