# Peer review of "Nitrous Oxide Adsorption and Decomposition on Zeolites and Zeolite-like Materials"

_molecules, 2022, doi:10.3390/molecules27020398_

Round 1

Reviewer 1 Report

The authors investigated the adsorption and decomposition of N2O on Zn-containing HZSM-5 zeolites and Titanosilicalites.

The mechanism of the reaction has been discussed on the basis of the spectroscopic data.

The results are very interesting and can be published on Molecules after minor reversions in the following:

  1. The pore structure of Zn-containing HZSM-5 zeolites and Titanosilicalites should be presented.
  2.  Except achieving the high efficient N2O decomposition,the selectivity to products should be concerned.
  3. Please give some comments on the function of the Zn and Cu in the metal-containing HZSM-5 in the decomposition of N2O.
  4. The Si/Ti ratio plays a great role in the decomposition of N2O over TS-1. For understanding the mechanism of the reaction, The Si NMR and NH3-TPD should be presented.

Author Response

First, we are very grateful to the reviewer for the valuable comments and we revised the manuscript by taking into account all these comments and recommendation. Below we present the comment and the response:

The authors investigated the adsorption and decomposition of N2O on Zn-containing HZSM-5 zeolites and Titanosilicalites. The mechanism of the reaction has been discussed on the basis of the spectroscopic data. The results are very interesting and can be published on Molecules after minor reversions in the following:

The pore structure of Zn-containing HZSM-5 zeolites and Titanosilicalites should be presented.

Response: The structure of all the zeolite samples studied in the paper, including the starting HZSM-5, dehydroxylated HZSM-5, Zn/HZSM-5, Cu/HZSM-5 and TS-1 samples with any Si/Ti ratio is the same MFI type. Dehydroxylation or modification with 3-5 wt.% Zn or Cu do not change the XRD patterns. Since they patterns are the same, we decided not to present them in the paper. This is included in the paper (page 9).

Except achieving the high efficient N2O decomposition, the selectivity to products should be concerned.

Response: The only products of N2O decomposition were N2 and O2. This is included in the manuscript (Materials and Methods section).

Please give some comments on the function of the Zn and Cu in the metal-containing HZSM-5 in the decomposition of N2O.

Response: The role of Zn and Cu in zeolite catalysts in N2O decomposition is discussed on page 5 (yellow text): “The presence of low-coordinated metal ions (zinc or copper) obviously should be considered as the pre-requisite for the efficient N2O decomposition. Thus, the spectroscopic and catalytic data indicate that strong electron-acceptor (low-coordinated) metal ions, which should be actually considered as Lewis acid-base pair sites containing a low-coordinated metal ion and an oxygen anion of the framework, are presumably the active centers responsible for the N2O decomposition on the modified zeolites.”

The Si/Ti ratio plays a great role in the decomposition of N2O over TS-1. For understanding the mechanism of the reaction, The Si NMR and NH3-TPD should be presented.

Response: We revised the discussion on page 9. However, we think that Si NMR data can hardly help to better understand the mechanism of N2O decomposition, as well as the ammonia TPD data. NH3-TPD can give only a profile of acidic sites in the catalysts without any differentiation between Broensted and Lewis (low-coordinated ions) acidity. The IR spectra of adsorbed N2O seem to be sufficient to support the hypothesis about the participation of coordinatively unsaturated titanium ions in the catalytic process.

Reviewer 2 Report

  1. The whole study of this manuscript is too simple. IR (Infrared Ray) was used too much, and other necessary catalyst characterization methods such as XRD (X-ray diffraction), TPD (Temperature programmed desorption) and SEM (Scanning electron microscope) were lacking.
  2. The experimental results in this paper are not as innovative as those in the most recent papers on the subject. The references in the manuscript are all too old, most of them published decades ago.
  3.  The English expression of the whole manuscript also needs to be improved.

Author Response

First, we are very grateful to the reviewer for the valuable comments and we revised the manuscript by taking into account all these comments and recommendation. Below we present the comment and the response:

The whole study of this manuscript is too simple. IR (Infrared Ray) was used too much, and other necessary catalyst characterization methods such as XRD (X-ray diffraction), TPD (Temperature programmed desorption) and SEM (Scanning electron microscope) were lacking.

Response: The structure of all the zeolite samples studied in the paper, including the starting HZSM-5, dehydroxylated HZSM-5, Zn/HZSM-5, Cu/HZSM-5 and TS-1 samples with any Si/Ti ratio is the same MFI type. Dehydroxylation or modification with 3-5 wt.% Zn or Cu do not change the XRD patterns. Since they patterns are the same, we decided not to present them in the paper. This is included in the paper (page 9). We think that ammonia TPD data can hardly help to better understand the mechanism of N2O decomposition. NH3-TPD can give only a profile of acidic sites in the catalysts without any differentiation between Broensted and Lewis (low-coordinated ions) acidity. The IR spectra of adsorbed N2O seem to be sufficient and most informative to support the hypothesis about the participation of coordinatively unsaturated titanium ions in the catalytic process. SEM studies are also not informative in the case of zeolites, since they are prepared using the same starting zeolite (HZSM-5) and the morphology and particle size and shape do not change after modifications via calcination and supporting Zn or Cu.

The experimental results in this paper are not as innovative as those in the most recent papers on the subject. The references in the manuscript are all too old, most of them published decades ago.

Response: We added and discussed a few very recent references related to the subject. Also, we removed unnecessary old publications from the list of references.

The English expression of the whole manuscript also needs to be improved.

Response: We checked the use of English and edited the manuscript.

Reviewer 3 Report

I have some, important doubts for the molecular mechanism presented on Scheme at line 163. In general, four-membered transition states (TS) are very tight and, in the consequence high energetical. More favoured are five-membered TSs. Within this type of TSs all angles are closer to typical sp2 configuration, and the energy of the activation are allways substantially lower. These issues regarding to the structural aspects of TSs within reactions with the participations of N2O were recently discussed in detail [Journal of Fluorine Chemistry, 160, 29-33 (2014) ]. This point of view should me mentioned in the text.

Table 2:
"W" parameter should be specified in the caption.

Key structures presented at Scheme within line 224 should be treatment with some caution. In general, this mechanism is rather not supported by experimental/quantumchemical data. This should be mentioned in the text.

Conclusions exhibit only a short constatation of facts. This paragraph should be substantially improved.

Author Response

First, we are very grateful to the reviewer for the valuable comments and we revised the manuscript by taking into account all these comments and recommendation. Below we present the comment and the response:

I have some, important doubts for the molecular mechanism presented on Scheme at line 163. In general, four-membered transition states (TS) are very tight and, in the consequence high energetical. More favoured are five-membered TSs. Within this type of TSs all angles are closer to typical sp2 configuration, and the energy of the activation are allways substantially lower. These issues regarding to the structural aspects of TSs within reactions with the participations of N2O were recently discussed in detail [Journal of Fluorine Chemistry, 160, 29-33 (2014) ]. This point of view should me mentioned in the text.

Response: We agree that the role of five-coordinated titanium cannot be ruled out. Therefore, we revised the discussion by adding the following: “It should be noted, however, that the presence of five-coordinated titanium ions can-not be excluded, since the XPS pattern represents a superposition of at least two, may-be three lines. The five-coordinated titanium ions are also coordinatively unsaturated and therefore, can take part in the reaction.” The reference proposed by the reviewer, however, is devoted to thermal (noncatalytic) decomposition of fluoronitroazoxy compounds and has no relation to the mechanisms of N2O decomposition on zeolites.

Table 2: "W" parameter should be specified in the caption.

Response: We specified the parameter.

Key structures presented at Scheme within line 224 should be treatment with some caution. In general, this mechanism is rather not supported by experimental/quantumchemical data. This should be mentioned in the text.

Response: We understand that our data do not provide a 100% probability of a certain mechanism, but the combination of spectroscopic (IR and XPS) and catalytic data agree with the first mechanism, whereas the choice between 4-coordinated and 5-coordinated titanium ions can be done only with the use of other quantitative methods.

Conclusions exhibit only a short constatation of facts. This paragraph should be substantially improved.

Response: We revised the conclusions.

Round 2

Reviewer 2 Report

The modified version already meets the standards and it is suggested to be accepted in present form.